# Visiting Peri-Urban Forestlands and Mountains during the COVID-19 Pandemic: Empirical Analysis on Effects of Land Use and Awareness of Visitors

Yuta Uchiyama [1] and Ryo Kohsaka [2,*]

1    Graduate School of Human Development and Environment, Kobe University, Kobe 657-8501, Japan;
yuta.uchiyama@landscape.kobe-u.ac.jp

2    Graduate School of Agricultural and Life Sciences, The University of Tokyo, Tokyo 113-8654, Japan

*    Correspondence: kohsaka@hotmail.com

**Abstract:** This research analyzed the status of visiting peri-urban forestlands and mountains during the first COVID-19 emergency period in Japan using a large-scale online questionnaire-based survey. We identified and examined the factors that correlated with visits to such areas, including respondents' social-economic attributes, environmental conditions (such as the land use patterns of their residential areas), and awareness of the functions of forestlands. The results suggest that environmental conditions are a major factor encouraging residents to visit peri-urban forestlands and mountains during the pandemic. Peri-urban areas with forestlands have such environmental conditions, and residents who visited peri-urban forestlands and mountains tended to live in peri-urban areas. Residents' expectations regarding forest functions were also strong factors influencing them to visit those places. Those who visited forests and mountains expected these areas to have mental health and educational functions. Especially, female respondents tended to be aware of forestlands as spaces for mental and physical relaxation, and respondents who have one or more children tended to be aware of the educational functions of forests. These findings imply that policy should consider the role of environmental conditions, awareness, and expectations about the function of forests and mountains, and prior interactions with nature in encouraging residents to visit such places for their health during the pandemic. These factors could also play a role in addressing the social and environmental disparities that exist between residents of different socio-economic statuses regarding access to nature. In future research, the detailed relationships between residents' environmental conditions and expectations/awareness of the functions of peri-urban forestlands and mountains need to be explored.

**Keywords:** peri-urban forestlands; mountains; COVID-19; land use; Japan

## 1. Introduction

Recently, access to and use of green areas have largely been affected by lockdowns and other control measures established in response to the ongoing COVID-19 pandemic [1–4]. The resultant unequal access to such areas, particularly in urban contexts, is one of the emerging injustices related to land. Specifically, existing studies e.g., [5,6] have extensively discussed the physical and mental effects of the limited access to green areas due to the pandemic. Preliminary studies have identified changes in the number of green area visitors. In addition to the overall number of visitors, certain studies have also analyzed their personal attributes [7]. Still, the target green areas are limited to parks, gardens, and agricultural lands. It is necessary to identify the status of visiting other types of green areas, such as peri-urban forestlands, which can provide spaces for relaxation, education, and other activities for visitors from urban and peri-urban areas. Furthermore, peri-urban forestlands are one of the essential components of biodiversity and ecosystem management [8–10].

The values of ecosystem services and their distribution are influenced by socio-economic factors [11]. In this circumstance, ecosystem service realization is needed to enhance citizens' awareness and facilitate participatory management [12]. Such awareness can be achieved using ecosystem services visualization, which policymakers and other experts can use to share the relevant scientific information with citizens [13]. Further, policymakers need to consider the tradeoff relationships between forest ecosystem services when formulating policy [14]. Indicator-based management is also an effective method for sharing the progress of policy implementations [15]. The main challenges of policymakers and experts are involving citizens in forest ecosystem services management and identifying stakeholders and their relationships.

To understand the status of forest ecosystem use by citizens, access and use of green areas and their values should be analyzed. Since before the pandemic, evidence of merits and issues surrounding forest ecosystem services have been provided [16]. For example, Kaźmierczak [17] highlights the direct and indirect contributions of green areas to citizens, while Wolch et al. [18] and Kabisch and Haase [19] suggest that environmental justice was one of the main concerns in issues related to the supply and demand of forest ecosystem services. Thus, to achieve fair and equitable sharing of ecosystem services, the distribution and usage trends of those services need to be identified to provide a basis for policymaking.

The impact of the pandemic varies among citizens with different socio-economic attributes [20,21]. For example, those belonging to racial minority groups and those with low income have been affected disproportionately by COVID-19 [20]. Regarding mental health, a greater impact of the pandemic on income is associated with increased depression and anxiety, while higher income is associated with better mental health conditions [22]. Similarly, the pandemic's effect on income levels also influences green area access—both physical and virtual—during the pandemic period [23]. A case study shows that low-income citizens, who are the most affected by COVID-19, have the least amount of nature-based environments nearby [24]. According to Pipitone and Jović [25], urban green areas are usually used by high-income citizens, and the pandemic has widened such socio-spatial disparities. Although the existing studies have detected this effect of income on green area use, this relationship is non-linear, and knowledge about the ideal green area in terms of health-related benefits is still lacking [26]. Furthermore, the influence of income on access and use of different types of green areas and the influence of environmental conditions such as land use in residential areas are not fully examined.

Considering the limitations of existing studies, the purpose of this research is to identify the status of visits to peri-urban forestlands and mountains during the first COVID-19 emergency period in Japan and to detect factors that correlate with such visits. This paper focused on residents' socio-economic attributes, land use, and awareness of the functions of forestlands because such factors may be correlated with green area use.

## 2. Materials and Methods

An online questionnaire survey was conducted to identify the status of visits to peri-urban forestlands and mountains. The survey period was from 31 July to 1 August 2020. Although there are discussions regarding the merit and demerit of online surveys e.g., [27], online surveys are still a useful method, especially when face-to-face surveys are not possible, as is the case during the COVID-19 pandemic [28]. The questionnaire was distributed to respondents in Aichi prefecture, Japan, two months after the first COVID-19 emergency period (16 April–14 May 2020) in Japan. During this period, the Japanese national government discouraged the residents from leaving their homes, especially to visit areas over prefectural borders. Although not following these recommendations did not result in penalties, there were significant effects on residents' behavior, particularly as Japanese society has relatively high peer pressure for following social norms and displaying a unified behavior. For example, it was reported that 37% of residents in Aichi did not visit green areas during the emergency period in previous studies on urban green areas [7]. Some residents were discouraged from even visiting hiking trails, although others visited them (Figure 1).

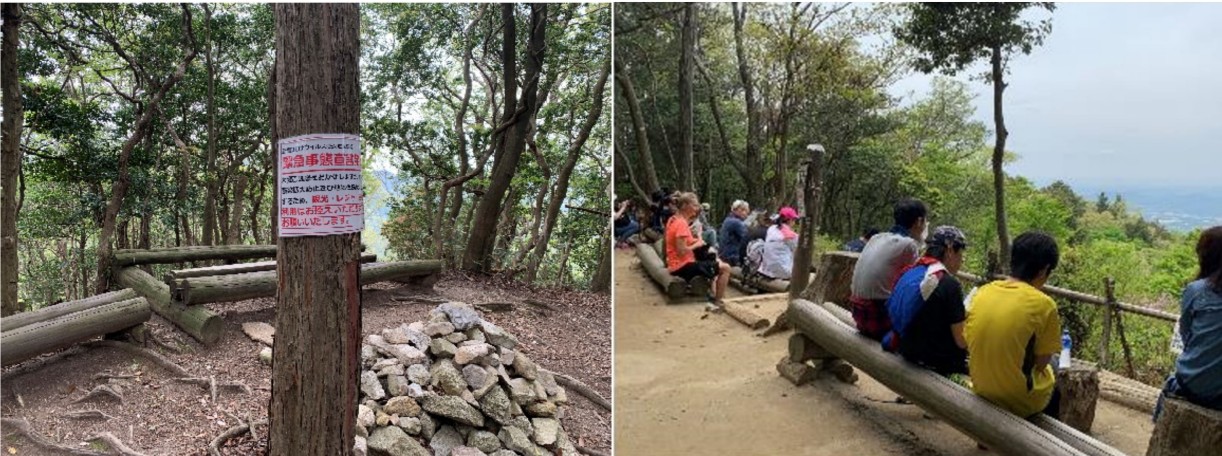

**Figure 1.** Signage discouraging people from visiting the hiking trail in Aichi Prefecture (**left**) and visitors of the same trail (**right**).

The research site, Aichi Prefecture, is the third largest metropolitan area (having the third largest population among other Japanese metropolises) with relatively good access to mountain areas and forestlands in Japan. Its capital is Nagoya City. The total number of respondents in this study was 1244. The female and male proportions were 47.6% and 52.4%, respectively. The proportion of over-60-years-old respondents was 36.6%; that of the five-year-interval age groups ranged between 7 and 11%, except the youngest age group of 20–24 years (2.9%).

The questionnaire consisted of questions posed to identify the status of visits to peri-urban forestlands and mountains at least once in the emergency period. The questions fell under four main categories as listed below. In the previous research, we analyzed the answers to questions about the status of visits to green areas, which were frequently visited by citizens during the emergency period [7]. In that research, the sample size of respondents who visited mountains (the main green area that was frequently visited) was too small to analyze (only 22 visitors), and mountain visitors were not analyzed.

- Socio-economic attributes: Sex (Male/Female), age, annual household income (1: <200, 2: 200–400, 3: 400–600, 4: 600–800, 5: 800–1000, 6: 1000–1200, 7: 1200–1500, 8: 1500–2000, 9: ≥2000 [Unit: 10 thousand JPY]), number of children in the household (>1/0);
- Environmental conditions: Zip-code district (the sizes of the areas and the ratios of land use categories in individual zip-code districts were computed and used in the analysis);
- Status of visits to peri-urban forestlands and mountains: Whether respondents visited mountains and forestlands during the emergency period (Answer: Yes/No);
- Awareness of the functions of forestlands (Respondents were asked about their pre- and post-emergency-period awareness of the functions of forestlands).

In order to compute the ratios of land use categories in zip-code districts, data from Japan Aerospace Exploration Agency's (JAXA) High-Resolution Land Use (2014–2016) survey (https://www.eorc.jaxa.jp/ALOS/en/dataset/lulc_e.htm [accessed on 28 May 2022]) were used. Based on the ratios of the land use categories and area sizes of the zip-code districts, the respondents' environmental conditions were analyzed. The resolution of the land use data was a 30 m square grid, and 10 land use categories were included in the data. Such high-resolution data were necessary to analyze the environmental conditions of zip-code districts with relatively smaller area sizes, which are located in urban areas. In order to analyze the overall environmental conditions of the individual zip-code districts, the ratios of urban areas, agricultural lands, and forestlands were computed.

Regarding respondents' awareness of forestland function, we assume that respondents who visited peri-urban forestlands and mountains have a certain level of awareness. To

analyze the characteristics of such awareness, we compared their awareness with that of respondents who did not visit these areas. We also performed a logistic regression analysis of their answers to the question related to forestland functions to identify the factors influencing their awareness.

## 3. Results

The results of the survey revealed that 212 (21%) respondents visited peri-urban forestlands and mountains during the emergency period. The reasons for visiting those places are shown in Figure 2. We have used the options of the reasons from the reference [29], which is research on forest recreation in the Japanese context. Considering the situation of the pandemic, we added "visiting safe place" as an option. More than 40% of the respondents who visited peri-urban forestlands and mountains visited them for relaxation. Feeling and touching nature was the second reason, and 20 to 25% of them answered with reasons including "to see a beautiful landscape," "to break from my daily routine," and "to visit a safe place." The latter reason ("to visit a safe place" [24%]) suggests that peri-urban forestlands and mountains were considered safe to a certain degree during the COVID-19 pandemic. Minor motivations for visiting included reasons directly related to physical activities, such as doing support exercises and recreational activities. The overall trend based on the given reasons for visiting mountains shows that the residents were motivated by their mental health.

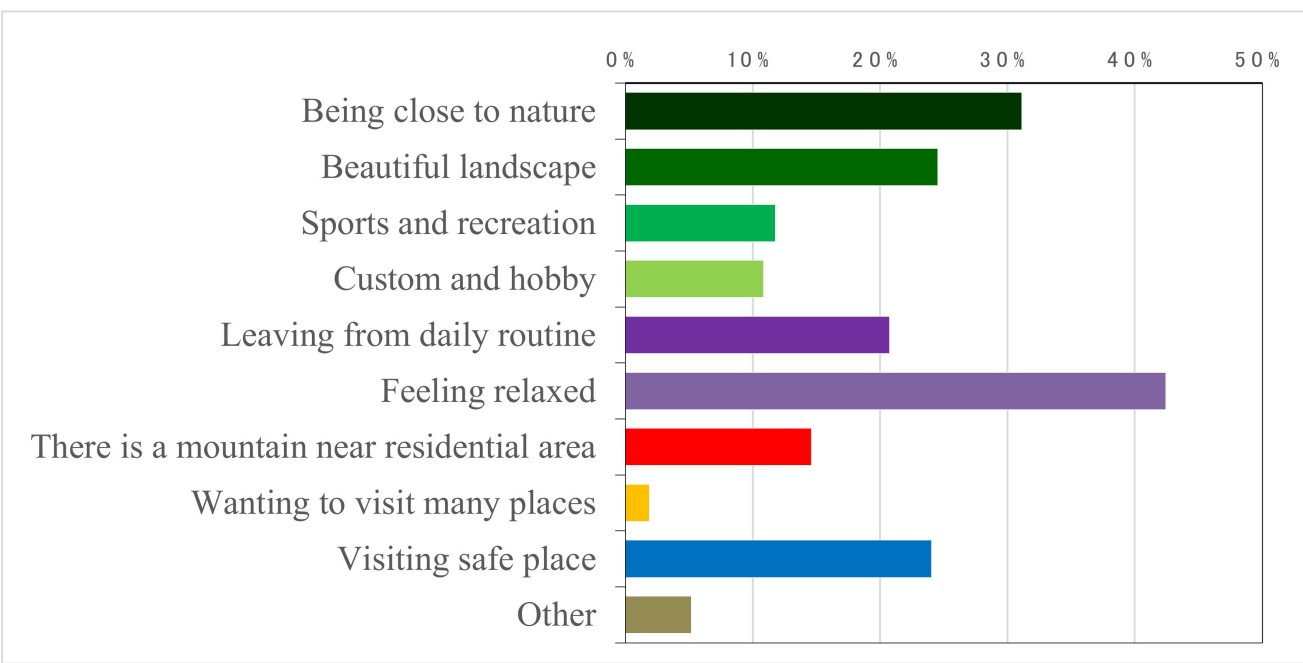

**Figure 2.** Reasons of visiting peri-urban forestlands and mountains.

Concerning the respondents' socio-economic attributes, there was no obvious difference between them except for gender. Figure 3 shows the male to female proportions of respondents who visited/did not visit peri-urban forestlands and mountains. The ratio of males who visited such places was high relative to that of females. Further, the over-60-years-old age group had the highest ratios of respondents in both categories (those who visited and those who did not visit), and thus it can be assumed that the older male respondents tended to visit those areas. When taken together, these results suggest that relatively older male respondents had a tendency to visit peri-urban forestlands and mountains.

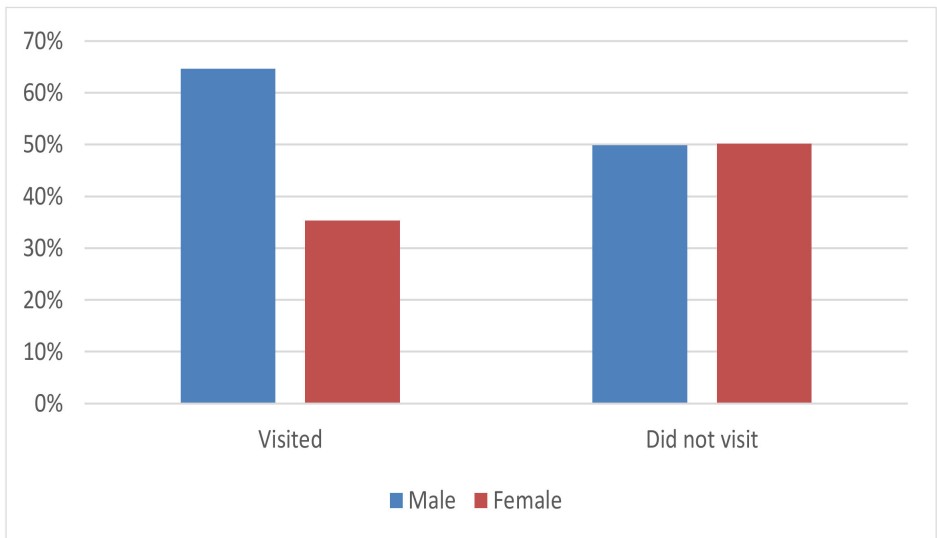

**Figure 3.** Ratios of respondents who visited and did not visit peri-urban forestlands and mountains by different gender groups.

Regarding residents' household income, an existing study i.e., [7] shows that residents with a higher household income had a tendency to visit green areas during the COVID-19 pandemic. However, Figure 4 indicates that in the current study, the difference in household income between the respondent groups (those who visited and those who did not visit forestlands and mountains) is unclear. That is, although relatively small differences in the ratios of respondents in individual household-income groups are evident between the two respondent groups (Figure 4), these differences are not as distinct as those shown in the existing study.

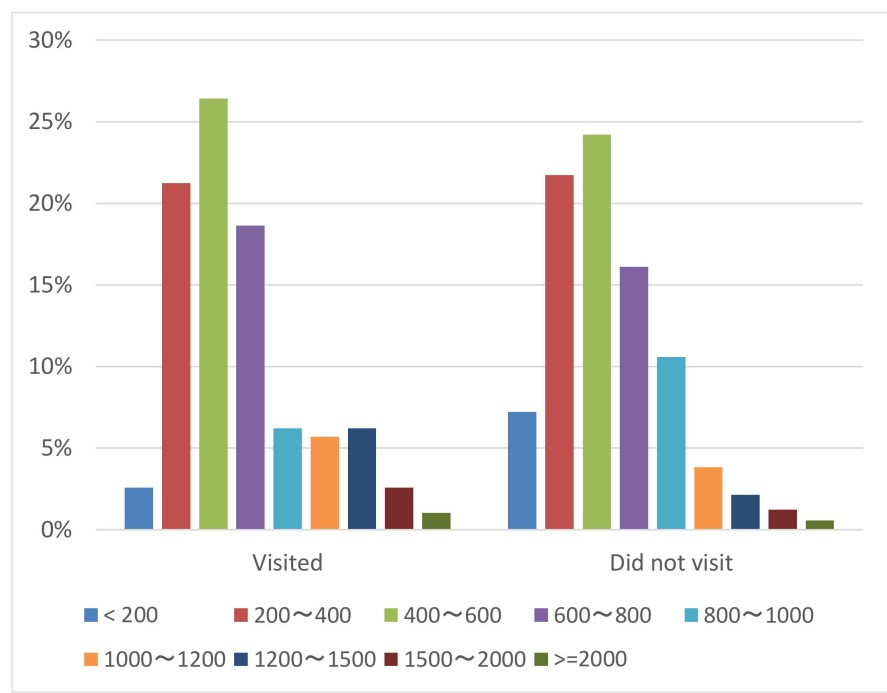

**Figure 4.** Ratios of respondents who visited and respondents who did not visit peri-urban forestlands and mountains by different household-income groups (Unit: million JPY).

Furthermore, the environmental conditions of respondents' residential areas were examined. It is revealed that there are statistically significant differences in the average ratio of forestland and area size of the zip-code district between the respondent groups who visited and those that did not visit peri-urban forestlands and mountains (Tables 1 and 2). Moreover, the average ratio of forestlands in the zip-code districts of respondents who visited such places is higher than that of those who did not visit ($t$-test, $p < 0.01$) (Table 1). This result suggests that the respondents who visited peri-urban forestlands and mountains can access forestlands in their residential areas relatively easily and might be more familiar with forest environments in their daily life. Additionally, it could be assumed that their residential places are not highly urbanized and not fully covered by built-up areas. Living in an environment that allows residents to easily access forestlands might be one of the factors encouraging residents to visit peri-urban forestlands and mountains.

**Table 1.** Ratios of forestlands in the zip-code districts of respondents who visited/did not visit mountains in the emergency period.

| Ratio of Forestland | Visited | Did Not Visit |
|---|---|---|
| Average (%) | 8.10 | 4.33 |
| Variance | 332.54 | 158.00 |
| Number of respondents | 212 | 1031 |
| Degree of freedom | 254 | |
| $t$ value | 2.87 | |
| $p$ value | 0.004426 | |

**Table 2.** Area sizes of the zip-code districts of respondents who visited/did not visit mountains in the emergency period.

| Area Sizes of the Zip-Code Districts | Visited | Did Not Visit |
|---|---|---|
| Average (ha) | 213.08 | 147.03 |
| Variance | 197,761.2 | 65,189.4 |
| Number of respondents | 212 | 1031 |
| Degree of freedom | 240 | |
| $t$ value | 2.09 | |
| $p$ value | 0.037421 | |

Regarding the area size of the zip-code district, respondents who visited forestlands and mountain areas lived in zip-code districts whose average area size was larger than that of respondents who did not visit ($t$-test, $p < 0.05$) (Table 2). Because zip-code area sizes of peri-urban areas are generally larger than those of urban areas, it can be assumed that the respondents who visited peri-urban forestlands and mountains live in peri-urban areas within the research site. In peri-urban areas, the residents may have relatively large residential land for their houses and gardens. In addition to access to forestlands, access to nature, such as plants in the garden, may be a condition that familiarizes the residents with nature and influences them to visit natural lands such as mountains.

In order to identify the distribution pattern of residential areas of respondents who visited peri-urban forestlands and mountains during the emergency period, the distribution of the respondents' zip-code districts was visualized; the results are shown in Figure 5. The figure also shows the zip-code districts of those who did not visit. As we mentioned previously, the smaller zip-code districts are located in urbanized areas. The central area of Nagoya City, which is the capital city of the research site, Aichi Prefecture, is indicated in Figure 5 with a red circle, and the sizes of zip-code areas are relatively small compared with those of the surrounding areas. As shown on the map, the zip-code districts of respondents who visited peri-urban forestlands and mountains (green-colored areas) are mainly located outside of the red circle, and their sizes are relatively large. This result may support the assumptions that we made based on the analysis of forestland ratios and sizes of zip-code

areas, showing that residential areas of respondents who visited peri-urban forestlands and mountains are concentrated in peri-urban areas, not in urbanized areas.

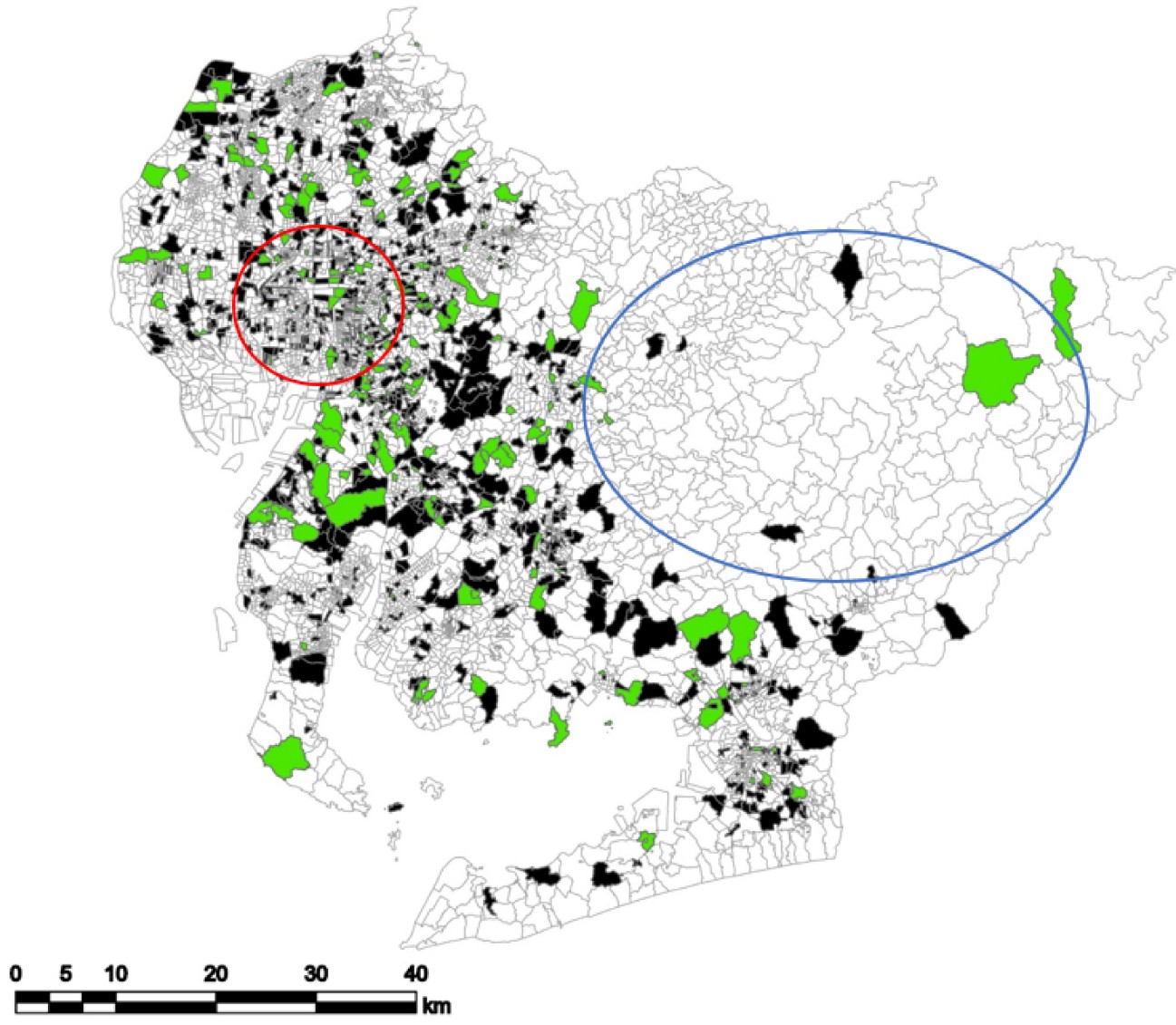

**Figure 5.** Distribution of zip-code districts of respondents who visited (green)/did not visit (black) forestlands and mountains during the emergency period in Aichi Prefecture. Note: Red circle shows the location of central urbanized area of the capital city (Nagoya City) of the prefecture and blue circle indicates the location of mountainous area with less population.

The analysis results of the respondents' environmental conditions imply that residents who do not have prior rich experiences of visiting forestlands and mountains might not feel encouraged to visit them during the pandemic.

Regarding awareness, we asked residents concerning their pre- and post-emergency-period awareness of forest functions. Specifically, we used the questionnaire to survey the residents' expectations concerning forest functions. We found that the ratios of respondents who expected certain functions differ between respondents who visited and those who did not visit peri-urban forestlands and mountains. As an overall trend, the difference between the residents' pre- and post-emergency-period expectations was relatively trivial. The different degrees of expectations between the two respondent groups were more evident among functions such as "providing a relaxing space," "providing an educational space," and "purification of air and reduction of noise." Before and after the emergency period, the ratios of respondents who visited forestlands and mountains and expected such

functions were higher than those of respondents who did not visit. After the period, the differences became larger for functions such as "providing a relaxing space" and "providing an educational space," as shown in Figure 6.

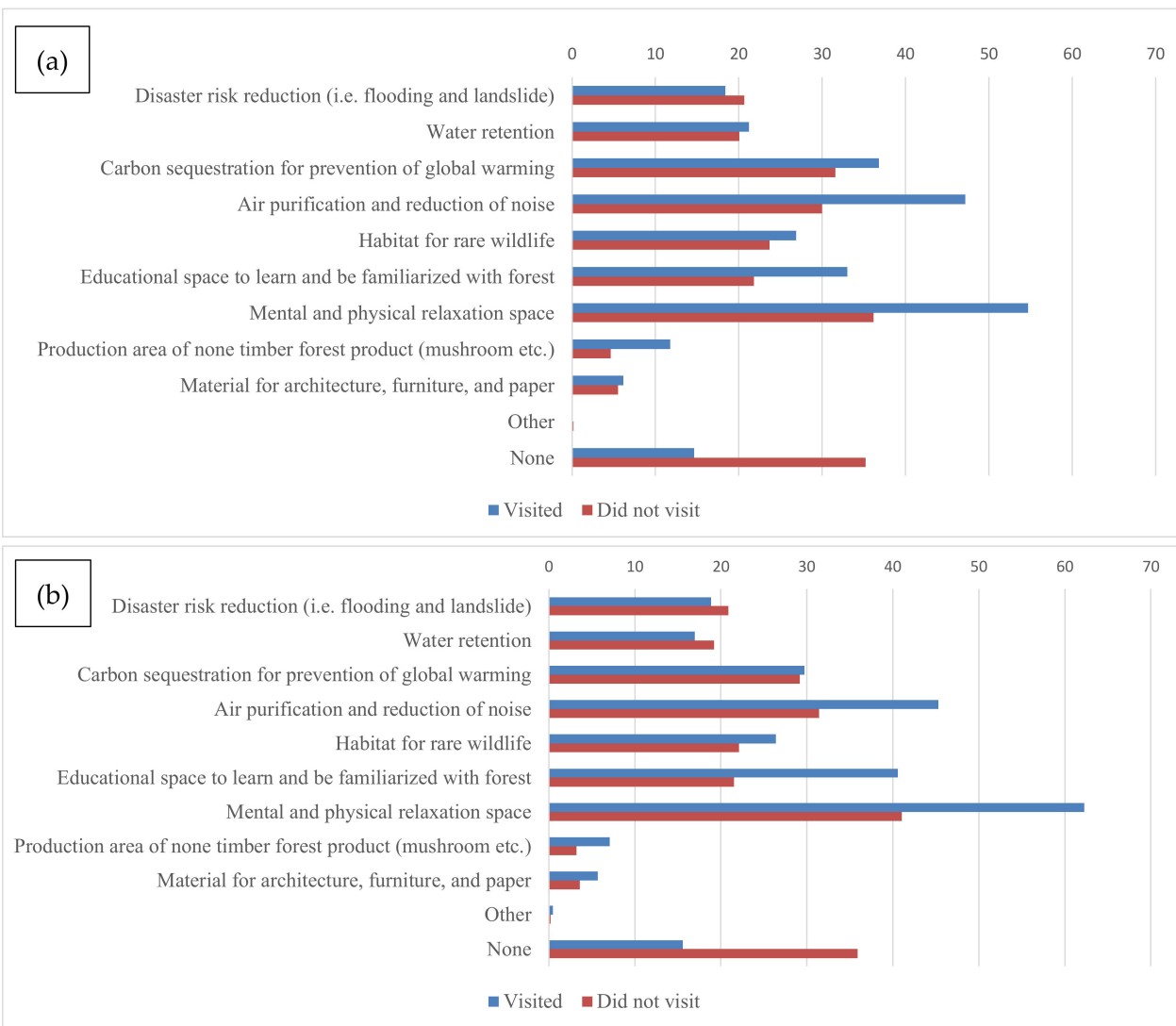

**Figure 6.** Awareness of the functions of forestlands and mountain areas (Respondents were asked about their awareness (**a**) before and (**b**) after the emergency period).

The results suggest that those who visited peri-urban forestlands and mountains highly expected these areas to serve mental- and education-related functions. This level of expectation may have encouraged the residents to visit those areas during the emergency period.

To identify the factors influencing awareness, we performed a binomial logistic regression analysis of the residents' responses. The results are shown in Tables 3 and 4. Statistically significant models were not detected for functions including "Disaster risk reduction [i.e., flooding and landslide]," "Habitat for rare wildlife," "Material for architecture, furniture, and paper," and "Other," meaning that awareness of these functions was not influenced by specific socio-economic attributes, environmental conditions, or whether they visited forestlands and mountains or not. The respondents' awareness of the other functions was influenced by these factors. Whether the respondents visited the forestlands and mountains or not was a statistically significant variable in most of the models for pre- and post-emergency-period awareness. This suggests that visiting such places can influence people's awareness of their functions. Several socio-economic environmental variables

were not statistically significant in influencing the respondents' pre-emergency-period awareness, even if they were significant for their awareness before the period. This implies that whether the respondents visited forestlands and mountains is a relatively strong factor influencing their awareness of the functions of these places, especially after the emergency period. Regarding other variables, household income was significant only in the model of awareness related to water retention. The respondents with higher income were more aware of that function. Being female was also a major influential factor alongside whether the respondents visited forestlands and mountains or not, and its coefficient was high and positive in the model of awareness related to mental and physical relaxation space. This suggests that the female respondents tended to be aware of the mental and physical relaxation function of forestlands and mountain areas. The coefficient of age was high in the model of awareness related to water retention. The older respondents tended to be more aware of that function. The number of children was significant in the model of awareness related to "educational space where ones can learn about and familiarize with forest." The respondents with one or more children were more aware of this function. Environmental conditions were significant only in the pre-emergency-period models. The forestland ratio had a negative coefficient in the carbon sequestration model (Table 3). This implies that the respondents living in zip-code districts with fewer forestlands (such as urbanized districts) tended to be aware of carbon sequestration. Concerning the size of the zip-code area, the results suggest that the respondents living in smaller zip-code districts, which are often located near or inside urban areas, tended to be more aware of the educational function of forestlands. Moreover, the results also show that the respondents living in larger zip-code districts, which are often located in rural areas, tended to be more aware of the production of non-timber products (mushroom, etc.) function (Table 3).

**Table 3.** Results of logistic regression analysis on the awareness of the functions of forestlands: before the emergency period.

| | Income | | | Sex | | | Age | | | Number of Children (>1 or 0) | | | Forestland Ratio | | | Zip Code Area size | | | Visited Forest or Not | | | Overall Model Test | | | | | |
|---|---|---|---|---|---|---|---|---|---|---|---|---|---|---|---|---|---|---|---|---|---|---|---|---|---|---|---|
| | Estimate | $p$ | Odds Ratio | Estimate | $p$ | Odds Ratio | Estimate | $p$ | Odds Ratio | Estimate | $p$ | Odds Ratio | Estimate | $p$ | Odds Ratio | Estimate | $p$ | Odds Ratio | Estimate | $p$ | Odds Ratio | AIC | Deviance | $R^2_{McF}$ | $\chi^2$ | df | $p$ |
| 2 | 0.10380 | 0.033 | 1.109 | −0.3207 | 0.074 | 0.726 | 0.02002 | 0.002 | 1.020 | | | | | | | | | | | | | 979 | 963 | 0.0243 | 24 | 7 | 0.001 |
| 3 | | | | 0.3161 | 0.038 | 1.372 | 0.0131 | 0.018 | 1.013 | | | | −0.014 | 0.057 | 0.986 | | | | 0.372 | 0.037 | 1.450 | 1204 | 1188 | 0.0141 | 17 | 7 | 0.017 |
| 4 | | | | | | | | | | | | | | | | | | | 0.681 | <0.001 | 1.975 | 1189 | 1173 | 0.0154 | 18.4 | 7 | 0.01 |
| 6 | | | | 0.3159 | 0.061 | 1.371 | −0.01214 | 0.045 | 0.988 | 0.699 | <0.001 | 2.012 | | | | −0.001 | 0.014 | 0.999 | 0.691 | <0.001 | 1.995 | 1018 | 1002 | 0.038 | 39.6 | 7 | <0.001 |
| 7 | | | | 0.4540 | 0.002 | 1.575 | | | | | | | | | | | | | 0.845 | <0.001 | 2.328 | 1255 | 1239 | 0.0266 | 33.9 | 7 | <0.001 |
| 8 | | | | | | | | | | | | | | | | 0.001 | 0.075 | 1.001 | 0.983 | 0.002 | 2.672 | 396 | 380 | 0.0594 | 24 | 7 | 0.001 |
| 11 | | | | −0.4172 | 0.007 | 0.659 | | | | | | | | | | | | | −1.135 | <0.001 | 0.322 | 1178 | 1162 | 0.0331 | 39.8 | 7 | <0.001 |

Note: Each model number shows the following functions of forestlands: 2. Water retention, 3. Carbon sequestration for prevention of global warming, 4. Air purification and reduction of noise, 6. Educational space to learn and be familiarized with forest, 7. Mental and physical relaxation space, 8. Production area of none timber forest product (mushroom etc.), 11. None. $R^2_{McF}$ is McFadden's $R^2$. The independent variable for sex is a categorical variable and its categories are male (1) and female (2). A positive estimated value of its coefficient indicates that females tend to be more aware of a function of forestlands than males. The table shows coefficients and other related values of significant variables ($p < 0.1$).

**Table 4.** Results of logistic regression analysis on the awareness of forestland functions: after the emergency period.

| | Income | | | Sex | | | Age | | | Number of Children (>1 or 0) | | | Forestland Ratio | | | Zip Code Area Size | | | Visited Forest or Not | | | Overall Model Test | | | | | |
|---|---|---|---|---|---|---|---|---|---|---|---|---|---|---|---|---|---|---|---|---|---|---|---|---|---|---|---|
| | Estimate | $p$ | Odds Ratio | Estimate | $p$ | Odds Ratio | Estimate | $p$ | Odds Ratio | Estimate | P | Odds Ratio | Estimate | $p$ | Odds Ratio | Estimate | $p$ | Odds Ratio | Estimate | $p$ | Odds Ratio | AIC | Deviance | $R^2_{McF}$ | $\chi^2$ | df | $p$ |
| 2 | 0.12992 | 0.009 | 1.139 | | | | 0.01982 | 0.003 | 1.020 | | | | | | | | | | −0.529 | 0.028 | 0.589 | 933 | 917 | 0.0224 | 21 | 7 | 0.004 |
| 4 | | | | | | | | | | | | | | | | | | | 0.634 | <0.001 | 1.885 | 1210 | 1194 | 0.0148 | 18 | 7 | 0.012 |
| 6 | | | | 0.3735 | 0.026 | 1.453 | | | | 0.579 | 0.002 | 1.784 | | | | | | | 1.092 | <0.001 | 2.980 | 1032 | 1016 | 0.0478 | 51 | 7 | <0.001 |
| 7 | | | | 0.4957 | <0.001 | 1.642 | | | | | | | | | | | | | 1.041 | <0.001 | 2.831 | 1274 | 1258 | 0.0369 | 48.2 | 7 | <0.001 |
| 8 | | | | | | | | | | | | | | | | | | | 0.856 | 0.017 | 2.353 | 329 | 313 | 0.0377 | 12.3 | 7 | 0.092 |
| 11 | | | | −0.4849 | 0.002 | 0.616 | | | | | | | | | | | | | −1.067 | <0.001 | 0.344 | 1187 | 1171 | 0.0316 | 38.2 | 7 | <0.001 |

Note: Each model number shows the following functions of forestlands: 2. Water retention, 4. Air purification and reduction of noise, 6. Educational space to learn and be familiarized with forest, 7. Mental and physical relaxation space, 8. Production area of none timber forest product (mushroom etc.), 11. None. $R^2_{McF}$ is McFadden's $R^2$. The independent variable for sex is a categorical variable and its categories are male (1) and female (2). A positive estimated value of its coefficient indicates that females tend to be more aware of a function of forestlands than males. The table shows coefficients and other related values of significant variables ($p < 0.1$).

## 4. Discussion and Conclusions

Our results suggest that residents who do not have rich experiences of visiting natural lands or access to such environments may lack the motivation to visit forestlands and mountains. This may mean that having such experiences and access to natural environments are essential factors influencing residents to visit forestlands and mountains to maintain good mental and physical health during the pandemic. Therefore, governments and local communities should establish policies and action plans to provide easy access to natural environments.

This study indicated that environmental conditions influence whether residents visit peri-urban forestlands and mountains during the pandemic. Specifically, peri-urban areas with forestlands have such environmental conditions, and the respondents who visited peri-urban forestlands and mountains tended to live in peri-urban areas.

Furthermore, the respondents' expectations concerning forest functions were also identified as factors that influenced residents to visit those places. The respondents who visited such areas had relatively high expectations that visiting forests would benefit their mental health and educational purposes. Environmental conditions may influence residents to have such expectations. The results of the logistic regression analysis show that visiting forestlands correlated with awareness of various types of forest functions, and the correlation between expectations, awareness, and whether or not the respondents visited forestlands became stronger after the emergency period. Regarding other correlations between respondents' attributes and forest function expectations, female respondents tended to be more aware of forestlands as mental and physical relaxation spaces, while respondents with one or more children tended to be aware of the educational functions of the forest. Although certain correlations between residents' environmental conditions and expectations or awareness of forestland functions were detected, in future research, these relationships need to be explored in more depth, considering that the value and meaning of forestlands differs from one country to another [30–32].

This paper contributes to the existing literature on environmental justice, specifically that related to the use of forest ecosystem services. Existing studies have examined the status of environmental justice in different socio-economic contexts [33,34], and the need to provide policy implications increased during the pandemic [35]. Additionally, extant studies e.g., [36] have also highlighted that environmental conditions are influential factors determining the behaviors of citizens in different regions. Nevertheless, studies focusing on environmental conditions are limited in this line of research. Future studies on forest visits and their environmental conditions can collaborate with the studies in different academic fields, such as public health [37], to analyze the effects of visiting forestlands on physical and mental health during the pandemic. Concerning the conservation of forest ecosystems and their biodiversity, the relationships between forest ecosystem services and biodiversity need to be identified to provide a basis for conservation policies and actions by citizens [38]. Since existing studies e.g., [39,40] have addressed the upscaling of valuations of forest ecosystem services, further research that considers visiting forestlands as use of forest ecosystem services also needs to be upscaled to provide policy implications for larger scales, such as country-level policy implications, considering ecosystem services and disservices [41].

"The extinction of experience" is a concern when discussing the essential role of having human–nature experiences in improving residents' understanding of biodiversity and ecosystem conservation. Although there are ongoing discussions about this concern in existing literature e.g., [42–44], such experience remains endangered, especially in urban areas and developed countries. An existing empirical analysis using large-scale official municipality data collected via a questionnaire-based residents' survey (cf. [45,46] for Sendai City) points to a decline in younger generations' interest in and recognition of the importance of human–nature interactions. During times of intense physical mobility limitations and mental pressures, experiences with nature can help improve residents' knowledge about the role of nature and increase their appreciation for conservation efforts.

In particular, options to address mental and physical health issues during the pandemic have been limited, especially for the vulnerable groups. Supporting, facilitating, and encouraging such groups to visit peri-urban forestlands and mountains may be a good alternative to both address their health issues and alleviate the social and environmental disparities between residents of differing socio-economic statuses regarding access to natural spaces. Further, discussions and future analyses around this issue could contribute to the debate related to the extinction of experiences and maldistribution of ecosystem services in social and environmental dimensions. Long-term analyses of similar issues in different world regions with or without continued mobility limitations are also needed.

**Author Contributions:** Conceptualization, Y.U. and R.K.; methodology, Y.U.; formal analysis, Y.U.; investigation, Y.U. and R.K.; writing-original draft preparation, Y.U.; writing-review and editing, Y.U. and R.K.; project administration, R.K.; funding acquisition, R.K. All authors have read and agreed to the published version of the manuscript.

**Funding:** This research was funded by the Japan Society for the Promotion of Science: JSPS KAKENHI Grant Numbers: JP17K02105; JP20K12398; 22H03852, JST RISTEX Grant Number JPMJRX20B3.

**Data Availability Statement:** The data presented in this study are available on request basis.

**Acknowledgments:** We would like to thank Yasushi Shoji who gave us valuable comments on the questionnaire.

**Conflicts of Interest:** The authors declare no conflict of interest.

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
