# Peer review of "Visiting Peri-Urban Forestlands and Mountains during the COVID-19 Pandemic: Empirical Analysis on Effects of Land Use and Awareness of Visitors"

_land, doi:10.3390/land11081194_

Round 1

Reviewer 1 Report

<General Evaluation>

This manuscript reported the effect of covid-19 pandemic on the visiting urban forests in a large-scale questionnaire survey in Japan. The study had selected the online questionnaire survey to identify the status of visiting peri-urban forestlands and mountains from 31 July to 1 August 2020 , that is two months after the first emergency period (16 April–14 May 2020) of COVID-19 in Aichi prefecture, Japan.

This ms had a merit in a viewpoint of question on the functional importance of Peri-Urban Forests and the related factors well documented. After first emergency period, what factors could visit the urban residents to the forests. However, I would like to suggest the structural problem in the title, analytic problem in designing the study.

This ms included the survey results in Aichi region, so the title can consider the regional case of study results.

Word of “environmental factors” could indicate the pollutants and meteorological factors, but this ms treated the land cover itself. So, Word of “land cover” could be appropriate for this study and could be included at the title.

<Detailed comments>

Figure 2 showed the percentage of 10 variables of reasons for visiting peri-urban forests, but could you ascertain the collinearity among 9 variables, “Being close to nature” and “Leaving from daily routine” could be associated with each other. So. Please clarify the collinearity of 9 variables, or suggest the related references.

Table 1 and 2 included the typo of number, so please correct

Figure 5 represents simple illustration on the urban and rural area, the comparison of figures or values between urban and rural areas could be better for reader.

Figure 6 represent the trivial difference before and after emergency period. Meaningful analysis could be presented with more rigorous analysis of results, what do you think about the appliance of electivity index before and after time.

L276~283, please write your comments on the data availability and acknowledgements, do not copy the guideline of the mdpi journal,

I would like to suggest the reanalyzing and rebuilding the study design for your valuable dataset and analytic tools.

Author Response

Thank you very much for your comments. Please kindly see the attachment.

Reviewer 2 Report

It is an interesting reserch  analyzed the status of visiting peri-urban forestlands and mountains during COVID 19 the  first emergency period in Japan via large-scale online questionnaire survey. The authors give in  proper way figures that give the essential roles of the experience in nature and  encourage residents to understand meaning of biodiversity and ecosystem conservation.  Although Table 1. with Ratios of forestlands in the zip-code districts of respondents who visited or did not visit mountains in the emergency period and Table 2 with the area sizes of the zip-code districts of respondents who visited / did not visit mountains in the emergency period have bad quality and appearance. They  are not necessary  to the manuscript, I propose to the authors to removed them and describe the data with text.

Author Response

Thank you very much for your commnets. Please kindly see the attachment.

Round 2

Reviewer 1 Report

Many parts had been developed and adjusted from the reviewer comments, and new insights were suggested. One figure was added on the ms. Overall, the merit of ms was enough to suggest the scientific soundness.